# In support of 2D:4D: More data exploring its conflicting results on handedness, sexual orientation and sex differences

**Denisa Cristina Lupu**[1], **Ignacio Monedero** [2], **Claudia Rodriguez-Ruiz**[1], **Miguel Pita**[1], **Enrique Turiegano**[1] *

**1** Departamento de Biología, Universidad Autónoma de Madrid, Madrid, España, **2** Departamento de Fisiología, Universidad Autónoma de Madrid, Madrid, España

* enrique.turiegano@uam.es

## Abstract

In the last few years, several studies have questioned the value of the second-to-fourth digit ratio (2D:4D) as a measure of exposure to sex hormones before birth. Controversy has also extended to the 2D:4D association with individual features previously related to this exposure such as handedness and sexual orientation. Given that it has been argued that sex differences in 2D:4D could be a consequence of body-size differences, we have tested in a large sample the allometric relationship between finger lengths and body size. Our results show that the association is either allometric or isometric, depending on the analyses performed. In any case, the deviation from isometry is not large enough to explain the typically observed sex difference in this trait. We have also tested the association between sexual orientation and 2D:4D, finding a relationship between 2D:4D and sexual orientation in men but not in women. We attribute this discordance with previously published meta-analysis to differences in genetic background, a variable that has gained relevance in recent years in studies involving 2D:4D. Finally, we did not find any relationship between 2D:4D and handedness, evaluated through self-reported preference and hand performance. Our main conclusion is that 2D:4D shows differences between sexes beyond their disparity in body size. In our opinion, 2D:4D can be used cautiously as an indicator of intrauterine exposure to sex hormones taking into account some considerations, such as analysing a very large sample and taking careful measurements of the ethnicity of the sample.

## Introduction

Adult females and males in most mammal species exhibit sex specific characteristics mainly determined by their exposure to sex hormones during embryonic development [1, 2], including behavioral features affected by the differential impact of these hormones on the developing nervous system [3, 4]. Humans are no exception, and both our anatomy and behavior are influenced by embryonic exposure to these hormones [5]. Some human characteristics that show sex differences related to fetal exposure to sex hormones are handedness [6, 7] and sexual orientation [8, 9]. However, the development of these characteristics, as the complex behaviors

**Funding:** ET- Spanish Ministry for Science and Innovation [grant numbers: BFU2010-10981-E; ECO2015-66281-P; PID2019-105895GB-I00]. (https://www.ciencia.gob.es/) ET- Ministry of Economics and Competitiveness [ECO2011-28750; ECO2012-33243] (https://portal.mineco.gob.es/) ET- Department of Biology (UAM) Research Funds [BIOUAM05-2019]. (https://www.uam.es/Ciencias/DBIO).

**Competing interests:** The authors have declared that no competing interests exist.

that they are, does not rely exclusively on endocrine factors, but also on genetic, environmental, and social aspects, and on the interaction between them all [8, 10]. There are also some features in adult individuals less dependent on sociocultural factors that have been used as indirect measures of the levels of sex hormones to which an individual has been exposed, like anogenital distance [11, 12] and spontaneous and click-evoked otoacoustic emission [13]. But none of them are as easy to measure and popular as the second-to-fourth digit ratio (2D:4D), the result of dividing the length of the second finger by the length of the fourth. Despite the popularity of 2D:4D as an indicator of sex hormone levels during fetal development [14], there has always been considerable controversy, strengthened in the last years [15, 16], as to whether it is a reliable measure. This has led to a great effort to attempt to replicate the initial results, review them, or carry out new studies to accumulate enough evidence to try to settle the question ([14, 17–24] among many others).

The controversy is mainly caused because the methods to directly measure fetal hormone levels lead to several problems. The most accurate way to measure prenatal exposure to testosterone is by sampling amniotic fluid. However, this is a hazardous procedure which could lead to miscarriage. Consequently, there are too scarce reliable data available. Some studies have explored the relationship between 2D:4D and testosterone in amniotic fluid despite this difficulty [18, 25–28]. These results suggest a correlation between 2D:4D and prenatal testosterone, but altogether they are not conclusive. In addition, amniocentesis procedures are carried out several weeks after the establishment of 2D:4D [27, 29] and at a time during development not so relevant for the shaping of the nervous system [5, 30, 31]. An alternative to amniocentesis has been measuring hormone levels in the umbilical blood right after birth [16, 28, 32, 33]. These studies show no correlation between 2D:4D and umbilical blood testosterone levels. However, testosterone levels measured from the umbilical cord at birth are substantially lower than those measured during pregnancy [30]. For that reason, it is not clear whether testosterone levels at birth are a good indicator of prenatal testosterone levels. A third possibility is to measure the level of sex hormones in the mother's blood. Again, the literature shows no clear correlation between maternal blood testosterone and newborn 2D:4D [28, 32]. Besides, it is difficult to obtain a definite conclusion about 2D:4D and fetal exposure to testosterone, given the differences between the mother's hormone levels and those the fetus is exposed to [34]. Many studies using these two collateral hormonal measures are aware of these limitations [16, 32].

On the other hand, there is indirect evidence of the connection between 2D:4D and hormone levels during fetal development. The differences between the sexes in this feature in many different populations, with men showing a smaller 2D:4D than women [35, 36], is the most difficult to justify with alternative explanations. Some authors plead that this difference could result from an allometric change in the shape of the hands [37, 38]. Another conflict with 2D:4D as an indicator of embryonic hormonal exposure is that it changes throughout life [39, 40]. However, Butovskaya et al. [21] have observed in a very large sample (> 7000) of different ethnicities and age ranges that sex difference in 2D:4D remains broadly stable with age and that this difference is not caused by allometry.

Additional evidence of the relationship between 2D:4D and fetal exposure to sex hormones derives from studies in humans affected by pathologies related to the levels or to sensitivity to sex hormones, as in the case of individuals suffering from congenital adrenal hypoplasia (CAH), Klinefelter syndrome or androgen insensitivity syndrome (summarized in [19, 35]). These studies also contribute to the current debate due to some conflicting results [22], but a recent meta-analysis has shown that 2D:4D is lower (denoting greater exposure to testosterone) in CAH populations compared to sex-matched controls [17]. Finally, many correlational and experimental studies indicate a relationship between 2D:4D and prenatal hormone levels

in amphibians [41–43], reptiles [44–48], birds [49–55], Artiodactyls [56], rodents [57–63] (but see [64]) and monkeys [65–71], although the direction of the difference between the sexes is not always the same observed in humans (lower 2D:4D in males).

Finally, some support for the relationship of 2D:4D with fetal exposure to sex hormones would come from its relationship to individual features which show sex differences. When these features are expressed very early in life, such as handedness [72] and, to a lesser degree, sexual orientation [8, 73], the effects of hormonal exposure on them are feasible. If these sex differences are at least partially dependent on sex hormone levels, some correlation of these features with 2D:4D should be expected within each sex [14]. However, the relationship between 2D:4D with these two individual features, handedness and sexual orientation, is far from being non-conflicting, with studies showing results in opposite directions in both cases.

Sexual orientation can be measured by focusing on sexual attraction, sexual behavior or self-identification [5, 8], the latter being the most frequently used way to evaluate it. In a large sample (more than 418,000 women and 378,000 men) from several surveys performed in western countries [74–78] a 1.95% of the sample considered themselves homosexual and 2.75% bisexual. Separating these data according to sex, 95.77% of men and 94.88% of women consider themselves heterosexual, while 1.60% of men and 3.80% of women consider themselves bisexual, and 2.63% of men and 1.33% of women consider themselves homosexual. The different frequency of homosexuals and bisexuals depending on the sex becomes accentuated when the subjects can choose between more than three options to define themselves. In such cases, in women the frequencies are progressively lower from completely heterosexual, the most frequent self-classification group, to completely homosexual. In men the less frequent choice is bisexual, being the highest proportions completely heterosexual and secondly completely homosexual [8]. The differences in sexual orientation between men and women include other issues such as the greater tendency to change sexual orientation throughout life in women [79] or the fact that sexual orientation is more linked to sexual arousal in men than in women [80, 81]. Different results link androgen exposure in embryonic development to this sex difference [82–84]. Considering only the studies focused on 2D:4D, a meta-analysis that includes 3121 men (48.2% homosexual) and 2707 women (37.5% homosexual) indicates that lesbians show more masculine 2D:4D than heterosexual women, while there is not a clear association between 2D:4D and sexual orientation in men [36]. Interestingly the association between 2D:4D and sexual orientation in men depends on ethnicity, finding an effect of different signs depending on the considered sample's geographic location [36]. Some authors have suggested that prenatal hormonal exposure is non-linearly associated with the development of sexual orientation in men, with some non-heterosexual men having been exposed to high androgen concentrations, while others were exposed to low concentrations [85–87]. Both low and high androgen signaling in male mice can lead to same-sex preferences [87]. On the other hand, some studies conducted after the abovementioned meta-analysis found a more feminine 2D:4D in homosexual men [88–91], especially when the approach to measure homosexuality allows to limit the study to strictly homosexual men [92]. In this regard, variables derived from self-reported physical measures related to androgen exposure during fetal development (2D:4D, height, strength) have been described as associated with men's sexual orientation in a very large population [85].

Handedness refers to the preference for using the right or the left hand to perform different tasks. Several methods classify individuals as left-handed or right-handed [93]. The most widely used method is to collect the self-reported preference from the subjects, either in general for any task or by detailing their particular preference for different tasks. Self-reported preference shows significant sex differences in some meta-analyses [6, 94]. In a sample of 2,396,170 people [6] 11.62% of men and a 9.53% of women were self-defined as left-handed.

Handedness can also be assessed implicitly, by measuring the ability of each hand to perform specific tasks. Interestingly, people who are better at performing tasks with their left hand tend to show a smaller difference in skills between their hands than right-handed individuals [95]. Measures of preference and performance of handedness correlate with each other [96–98], although measures of preference usually present a bimodal distribution (left-handed versus right-handed) that is not observed when performance is measured [93].

Handedness is determined before birth [72] and seems to be related to androgen exposure during development (reviewed in [7]). Initial studies showed that women with CAH were more frequently left-handed [99], but further meta-analysis did not observe this effect [100]. On the other side, some studies which directly analyse the relationship between the concentration of sex hormones in amniotic fluid and handedness indicate that high prenatal exposure to androgens in girls predisposes to less lateralization [7]. Regarding the relationship of handedness with 2D:4D, a recent meta-analysis including more than 220,000 subjects of 107 different samples verifies the existence of a significant relationship, although relatively weak and inconsistent [24]. The meta-analysis shows a significant correlation between a lower right hand 2D:4D (more masculine) and the tendency to be non-right-handed. This trend is maintained when the sample consists only of women, or in studies with non-dichotomous measures of handedness. There is also a relationship between being non-right handed and low values in the difference between the 2D:4D of both hands [100], which is also associated with fetal exposure to higher testosterone levels. This is observed when the sample consists only of men or women and with continuous or discontinuous measures of handedness. The inconsistency is explained by an association between left-handedness and a higher (more feminine) 2D:4D in the left hand, which also comes out in samples of men and women analysed separately and considering continuous and discontinuous measures of handedness. Thus, a lower 2D:4D in the right hands and a higher 2D:4D in the left hands seems to be associated with left-handedness. Interestingly, the meta-analysis does not compare handedness measures of preference versus performance, as the latter are less frequently used. Previous works focused on hand performance frequently show an association between 2D:4D and being left-handed, some of them linking handedness with the difference in 2D:4D between hands [101, 102], while some others link it with right 2D:4D [103, 104] or even with left 2D:4D [104]. However, some studies found no association between the differences between hand performance and any of these values [105].

Also many studies analysed the relationship between handedness and sexual orientation beyond its relationship with 2D:4D. In fact, many research efforts conducted to analyse biological processes contributing to sexual orientation development focus in this feature as marker of cerebral lateralization determined by endocrine, genetic and immunological mechanisms [8, 106, 107]. The proportion of self-reported non-right-handed is higher among homosexuals, although the relationship seems stronger between women according to a meta-analysis made in 2000 including more than 23,000 subjects [108]. Subsequent studies indicate that in women the association between non-right handedness and homosexuality lies mainly in a higher proportion of mixed handed [74, 78, 109, 110]. In men the relationship of non-right handedness with homosexuality is not that clear in studies performed after the meta-analysis. It is supported by some [74, 110–112] but without detecting such relationship in some others [78, 109, 113, 114]. Some other studies have found homosexual men show an extreme preference for using only the right or the left hand, contrary to what is observed in women [115, 116]. Finally, there are also studies in which there is no relationship between these variables in any sex [117] (but N = 167). One of these works [118] (N = 1283) also found no relationship between homosexuality and non-right-handedness, although it did find a relationship between asexuality and non-right-handedness.

Over several years we have generated a considerable amount of data that allow us to analyse all these problematic issues related to 2D:4D in a relatively large sample of young participants. In this population we studied whether the difference in 2D:4D between men and women is due to allometric issues, and also the relationship between 2D:4D and handedness measured with two different methods and the relationship between 2D:4D and sexual orientation. Some of these data have been previously published [119, 120]. Reanalysing previously published data has the advantage that it prevents intentional data selection [121]. By analysing this sample we intend to contribute to the information available on these three controversial points. In addition, we have measured handedness based on performance, not just preference, which we believe has never been used for this purpose before.

As in a recent previous report [21], we did not expect to find strong evidence for an allometric cause behind the sex difference in 2D:4D. We expected to replicate the trends described above regarding the relationship between 2D:4D and sexual orientation and handedness. We also expected that the associations will be easily detected when both individual features are measured with a scale, given that particularly strong relationships appear when more detailed categorical classifications are used (such as mixed handed, or completely homosexual). Some of the studies mentioned above propose that these strong links are behind the observed results when considering only two larger categories (non-right handed non-heterosexual). Finally, we also tested the relationship between handedness and sexual orientation, even when it is beyond an expected link with 2D:4D.

However our results do not fully support the previously described relationships between 2D:4D and handedness or sexual orientation, increasing the need to pay more attention to the genetic background of the samples. Regarding the association between 2D and 4D, our results indicate that the relationship between the fingers seems to be on the border between isometry and allometry. Moreover, it seems to be different in men and women. Altogether, in this study we contribute to the debate with data not collected to test this question specifically, which minimizes self-selection sampling issues and observer bias problems.

## Materials and methods

The sample includes 935 men and 900 women who have participated in different experiments since 2010, some whose results have been published and others are pending publication. The sample includes right hand scans of 147 men from Sanchez-Pages and Turiegano (2010) [119] and the right and left hand scans of 474 women and 328 men from Rodriguez-Ruiz et al. (2019) [120] (with the exception of 37 women hand pairs whose original images could not be found). We included data already published in the study because this makes data dredging impossible in that subsample [121]. The sample also lacks data from one hand in some subjects, because the excluded scans did not have sufficient quality or because the same hand was scanned twice by mistake. Individuals included in the sample ethnically considered themselves as white asked through an open-ended question. Their nationality is mainly Spanish (92.53%), with a 5.02% from other European countries (UK, Germany, Russia, Romania, Italy. . ..), a 1.74% from South American Countries (Argentina, Chile, Venezuela. . ..), a 0.38% from North African Countries (Morocco, Tunisia, Egypt) and a 0.33% from Middle East Countries (Israel, Lebanon, Iran. . .).

Data analysed were obtained in different experiments. All the experimental procedures received the written consent of the Comité de Ética de la Investigación (CEI) (Research Ethics Committee) of the Universidad Autónoma de Madrid (Permit numbers: 25–575, 27–642, 55–997, 73–1319, 75–1364, 94–1722). Each participant signed an informed consent form granting authorisation for the use of the data for research purposes.

Finger length was measured independently by two investigators using the TPS morphometric analysis software package TPS version 2.16 (by F.J. Rohlf; obtained from http://life.bio. sunysb.edu/ee/rohlf/software.html). The fingers were measured from the top of the digit to the center of the flexion crease proximal to the palm [119]. Investigators were blinded to the sex of the participant to avoid observer bias. Image samples were randomly ordered independently for each researcher to avoid problems due to observer drift. Intraobserver correlations calculated for a sample of 200 fingers repeated measures were (both) greater than r = 0.975. The correlation between the measures taken by the two researchers was very high ($r_{7006}$ = 0.992, p <0.001). The length values of each finger used in the analyses are the average of these two measures. The 2D:4D index of each hand is calculated by dividing the length of the second finger by the fourth. Table 1 shows the values obtained for each finger, as well as the 2D:4D index of each hand for the whole sample in men and women. 2D:4D values for both hands follow a normal distribution (Right K-S = 0.021 p = 0.050; Left K-S = 0.018, p> 0.200). The correlation between the 2D:4D of the right and left hands was significant and positive ($r_{1668}$ = 0.722, p <0.001). The correlation between the currently measured 2D:4D and the published 2D:4D in the published data subsample is very high for both the right ($r_{866}$ = 0.902, p <0.001) and the left hand ($r_{797}$ = 0.930, p <0.001).

To check whether sexual dimorphism in 2D:4D is a consequence of an allometric shift in the shape of the fingers, we performed two types of linear regressions. First, we carried out ordinary least squares regressions (OLS) between the 2D:4D index and the average size of the two fingers [121]. If there is finger isometry, a slope of 0 and an intercept of 1 are expected. This regression is relevant because the variable widely used in previous literature is 2D:4D. Second, we also performed reduced major axis regressions (RMA) between the lengths of the second and fourth fingers, assuming an equal proportion of noise in both lengths [122]. In case of isometry the expected slope would be 1 and the expected intercept would be 0. This regression is more relevant to determine whether the size relationships between the fingers are isometric or not [121]. We did these analyses separately for the right and the left hands on the whole sample and in the previously published data sample, separately in men and women and including them all together.

Sexual orientation was assessed using two different methodologies. In some experiments (502 women, 389 men) it was collected by asking the participants to choose one of three possible categories to define their sexual orientation: homosexual, heterosexual or bisexual. In another part of the sample (443 women, 341 men) they were asked to place themselves at the considered point of a straight line that went from heterosexual to homosexual. This variable does not present a normal distribution (K-S = 0.230, p <0.001). 1.95% of men and 0.78% of women did not answer this question when asked. The values obtained for men and women in both measures of sexual orientation are shown in Table 1. For a small part of the sample (52 women and 26 men) both measurements were carried out. In this sample, the values of the continuous variable ranged from 0.045 to 0.450 for people who considered themselves as homosexual (N = 4), from 0.459 to 0.757 for those who considered themselves as bisexual (N = 8) and from 0.676 to 1.000 for those who considered themselves as heterosexual (N = 66). The differences between the three groups for the continuous measure of sexual orientation were significant ($F_{2.75}$ = 154.260, p <0.001). In order to use the entire sample to perform some analyses, we generated a discrete sexual orientation variable for those participants in which we only had the continuous measure, which we have called ESOC (estimated sexual orientation category). Based on the data obtained in the population for which we had both measures, we decided to consider those subjects with values greater than 0.7 as heterosexual, those with values below 0.3 as homosexual, and those with values between 0.3 and 0.7 as bisexual. This method considers a range of equal length to categorize participants as heterosexual or

**Table 1. Summary statistics.**

|  | Women | Men |  |
|---|---|---|---|
| Right 2D:4D | 0.975 ± 0.001 (895) | 0.962 ± 0.001 (934) | $t_{1927}$ = 9.065, p<0.001, d = 0.424 |
| Left 2D:4D | 0.973 ± 0.001 (899) | 0.965 ± 0.001 (775) | $t_{1672}$ = 9.065, p<0.001, d = 0.265 |
| Right 2D length | 6.868 ± 0.013 (896) | 7.478 ± 0.014 (934) | $t_{1927}$ = -31.648, p<0.001, d = 1.481 |
| Right 4D lenght | 7.044 ± 0.014 (895) | 7.779 ± 0.015 (934) | $t_{1672}$ = -36.576, p<0.001, d = 1.713 |
| Left 2D length | 6.838 ± 0.013 (899) | 7.496 ± 0.016 (775) | $t_{1927}$ = -32.157, p<0.001, d = 1.571 |
| Left 4D lenght | 7.031 ± 0.014 (899) | 7.777 ± 0.017 (775) | $t_{1672}$ = -34.984, p<0.001, d = 1.706 |
| Right Fingers Av. | 6.956 ± 0.013 (895) | 7.628 ± 0.014 (934) | $t_{1927}$ = -35.654, p<0.001, d = 1.669 |
| Left Fingers Av. | 6.935 ± 0.013 (899) | 7.636 ± 0.016 (775) | $t_{1672}$ = -34.990, p<0.001, d = 1.708 |
| Age (yr) | 20.971 ± 0.082 (895) | 21.454 ± 0.127 (934) | U = 383911.0, p = 0.002, d = 0.149 |
| SexOrCont | 0.828 ± 0.010 (443) | 0.800 ± 0.016 (341) | U = 73741.0, p = 0.561, d = 0.112 |
| SexOrCateg | Het: 90.6%, Bi: 6.6%, Hom:2.8% (502) | Het: 88.7%, Bi: 3.1%, Hom:5.2% (389) | $\chi^2$ = 17.926, p<0.001 |
| dHGS | 7.165 ± 0.518 $10^{-2}$ (417) | 5.982 ± 0.401 $10^{-2}$ (774) | U = 148936.5, p = 0.028, d = 0.109 |
| Right-handed | 91.8% (417) | 87.9% (776) | $\chi^2$ = 4.441, p = 0.035 |

For continuous variables, the mean ± SEM is indicated. For categorical variables the percentage of sample in each category is indicated. The total N appears in parentheses. Finger lengths are provided in centimeters. Het: heterosexual, Hom: homosexual, Bi: bisexual. In continuous variables d is Cohen's d for sex difference. Effect size for Cohen's d: negligible (d<0.20), small (0.20≤d<0.50), medium (0.50≤d≤0.80) and large (d>0.80).

homosexual (0.3 in this case), a common consideration when categorizing continuous variables that measure sexual orientation [8]. Our method of classification maximizes the number of properly classified subjects, as ESOC correctly classifies 96.2% of the sample whose sexual orientation was measured with both continuous and categorical measurements. ESOC also shows significant differences ($\chi^2$ = 67.636, p <0.001) in the distribution of sexual orientations between men (Hom: 10.7%, Bi: 5.0%, Het: 84.4%) and women (Hom: 2.7%, Bi: 13.1%, Het: 84.2%).

We use two different methods to measure handedness. First, we use the stated preference on how the subjects consider themselves (right-handed, left-handed or ambidextrous). Second, we use a performance measure of handedness (dHGS), the difference in hand grip strength between the hands divided by the average strength (equivalent to the lateral quotient from [96, 97]). HGS was measured twice in each hand alternating measurements, following the instructions of the manufacturer of the hand-grip dynamometer used (Saehan SH5001). The maximum HGS for each hand was used in the analyses. dHGS does not follow a normal distribution (K-S = 0.082, p <0.001). The values for self-considered preference and dHGS in women and men are shown in Table 1. Differences in dHGS between subjects who consider themselves right handed (7.021 ± 0.303 $10^{-2}$) and non-right handed (-5.436 ± 0.108 $10^{-2}$) are significant (U = 25587.5, $N_1$ = 1063, $N_2$ = 128, p <0.001).

To test the validity of dHGS, we compared it with two other measures of handedness in a small subsample (127 women, 100 men). First, we used a 7-item version test [123] of the Edinburgh Handedness Inventory (EHI) translated into Spanish [124], but considering only three possible categories for each task. We compute +1 if the right hand is used for a certain task, 0 if both hands are used and -1 if the left hand is used. EHI values are not normally distributed (K-S test = 0.087; p <0.001). Second, we use a hand performance measure that compares the ability to draw a contour with the right and left hand (ErrRL). We compare the errors made in following the contour between one hand and the other by measuring the area that remains between the pen drawing and the original silhouette (ErrRL = right error—left error). We measure the areas using a macro in ImageJ [125]. ErrRL does not follow a normal distribution

(K-S test: 0.405; p <0.001). In this subsample we found no significant differences between the proportion of right-handed men and women, and their values are similar to those reported in the literature (87.1% men, 92.2% women). The distribution of dHGS in the subsample is normal (K-S test = 0.043, p> 0.200), and there are significant differences between right-handed and not right-handed in ErrRL (Mean ± SD; Right: -299.743 ± 302.664; Left: 129.500 ± 218.038; U = 345.00, p < .001), dHGS (Right: 0.088 ± .103; Left: -0.050 ± .122; U = 742.00, p <0.001) and EHI (Right: 7.471 ± 1.298; Left: -4.100 ± 4.179; U = 125.50, p <0.001). The correlations between dHGS and ErrRL ($\rho_{229}$ = -0.235; p <0.001) and between dHGS and EHI ($\rho_{229}$ = 0.287; p <0.001) are significant and in the expected direction. ErrRL and EHI also correlate in the expected direction ($\rho_{229}$ = -0.422; p <0.001).

Statistical analysis was performed using PAST 4.03 [126] and SPSS 15.0.1. The comparison between regression coefficients and correlation parameters was calculated following the method described by Zar (1998) [127]. These tests were preferred over including Sex and Size as independent variables in a GLM, in order to avoid collinearity problems. Parametric and non-parametric tests were performed according to the normality of the variables considered.

## Results

According to OLS regressions (Table 2), the relationships between the fingers appear to be mostly isometric in men and women in both hands and in both the entire sample and the subsample of published data. The slope is not significantly different from 0 except in the regression fitted for the right-hand data for the men in the full sample. The slopes are not significantly different between women and men neither in the complete sample (right: $F_{2,1791}$ = 3.836, p = 0.050; Left: $F_{2,1672}$ = 1.472, p = 0.225) nor in the subsample (right: $F_{2,943}$ = 0.116, p = 0.734; Left: $F_{2,799}$ = 0.731, p = 0.393). The intercept is not significantly different from 0 except in the regression adjusted to the right-hand data of the women in the complete sample. However, when we simultaneously consider both men and women in most of the adjusted

**Table 2. OLS regression of 2D:4D onto mean fingers length.**

| | | | WOMEN | | MEN | | ALL | |
|---|---|---|---|---|---|---|---|---|
| | | | Coeff.±SEM | 95% BCI | Coeff.±SEM | 95% BCI | Coeff.±SEM | 95% BCI |
| RIGHT HAND | Full sample | SLOPE | 0.002[NS]±0.003 | (-0.003, 0.007) | -0.005*±0.002 | (-0.010, 0.000) | -0.009**±0.001 | (-0.012, -0.005) |
| | | INTERCEPT | 0.959+±0.020 | (0.920, 0.999) | 1.000[NS]±0.019 | (0.963, 1.038) | 1.036++±0.011 | (1.016, 1.058) |
| | | | $R^2$ = 0.001 | | $R^2$ = 0.004 | | $R^2$ = 0.022 | |
| | Sub-sample | SLOPE | -0.002[NS]±0.004 | (-0.010, 0.006) | -0.004[NS]±0.004 | (-0.011, 0.003) | -0.010**±0.002 | (-0.014, -0.006) |
| | | INTERCEPT | 0.988[NS]±0.028 | (0.933, 1.043) | 0.990[NS]±0.027 | (0.938, 1.046) | 1.040+±0.015 | (1.010, 1.071) |
| | | | $R^2$ = 0.000 | | $R^2$ = 0.002 | | $R^2$ = 0.023 | |
| LEFT HAND | Full sample | SLOPE | 0.001[NS]±0.003 | (-0.004, 0.006) | -0.004[NS]±0.003 | (-0.009, 0.002) | -0.006**±0.001 | (-0.009, -0.003) |
| | | INTERCEPT | 0.967[NS]±0.019 | (0.934, 1.002) | 0.994[NS]±0.020 | (0.953, 1.034) | 1.013[NS]±0.011 | (0.993, 1.032) |
| | | | $R^2$ = 0.000 | | $R^2$ = 0.003 | | $R^2$ = 0.010 | |
| | Sub-sample | SLOPE | -0.003[NS]±0.004 | (-0.011, 0.004) | -0.008[NS]±0.004 | (-0.017, 0.001) | -0.007**±0.002 | (-0.012, -0.003) |
| | | INTERCEPT | 0.995[NS]±0.027 | (0.945, 1.046) | 1.028[NS]±0.033 | (0.961, 1.092) | 1.013[NS]±0.016 | (0.992, 1.053) |
| | | | $R^2$ = 0.001 | | $R^2$ = 0.011 | | $R^2$ = 0.014 | |

We used t-test (2-tailed) to test if the slopes are different from 0 (** p<0.001; * p<0.05; [NS] p>0.05) and if intercepts are different from 1 (++ p<0.001; + p<0.05; [NS] p>0.05). The common regression slopes built from the full sample regression of women and men are -0.00168 for the right hand and -0.00159 for the left hand. Common regression intercepts are 0.98081 for the right hand and 0.98067 for the left hand. N for right hand consisted of 895 females and 934 males, while N for the left hand was 899 females and 775 males. The subsample of published data includes 474 males and 471 females for the right hand and 327 males and 474 females for the left hand. BCI: Bootstrap confidence intervals. The 95% bootstrap confidence intervals were computed with 1999 rearrangements.

regression there are significant differences from the values associated to isometry for both slopes and intercepts (Table 2).

We analysed the residuals obtained with the regressions including simultaneously men and women for the complete sample. The residuals from both the right hand (K-S = 0.019, p = 0.126) and the left hand (K-S = 0.021, p = 0.084) follow a normal distribution. There is no significant correlation between residuals and the average finger length for both the right hand ($r_{1829}$ = 0.000, p = 1.000) and the left hand ($r_{1674}$ = 0.000, p = 1.000). There are significant differences ($t_{1827}$ = 4.881, p <0.001) in the residual values for the lines adjusted to right hand data between men (-3.606 ± 1.034 $10^{-3}$) and women (3.763 ± 1.102 $10^{-3}$). The residuals obtained in the left hand for men (-2.272 ± 1.151 $10^{-3}$) and women (1.958 ± 1.062 $10^{-3}$) were also significantly different ($t_{1672}$ = 2.703, p = 0.007). Women tend to have a 2D:4D larger than expected according to the size of their fingers, while in men 2D:4D is smaller than expected due to their size.

In RMA regressions (Table 3) we observed that in women the slope and intercept values indicate isometry. In the case of men, the intercept of none of the adjusted lines was significantly different from zero. However, the slopes of all regressions performed with men finger lengths were significantly greater than 1. Nevertheless, when comparing the slopes between women and men there is only a sex difference for the right-hand data in the entire sample ($\chi^2$ = 6.308, p = 0.012), whereas this is not the case for the left-hand data ($\chi^2$ = 2.261, p = 0.132) nor for the subsample (right: $\chi^2$ = 0.435, p = 0.509; Left: $\chi^2$ = 0.986, p = 0.321). When we include data for both sexes jointly, the slope and intercept were significantly different from the values associated with isometry in all cases, with finger 4 being larger than finger 2.

Residuals for the regressions including data for women and men simultaneously are normally distributed for both the right hand (K-S = 0.013, p> 0.200) and the left hand (K-S = 0.016, p> 0.200). There are significant differences ($t_{1827}$ = -5.243, p <0.001) in the residuals obtained for the right hand between women (-3.185 ± 0.823 $10^{-2}$) and men (3.052 ± 0.857 $10^{-2}$). The differences are also significant ($t_{1672}$ = -3.019, p = 0.003) in the

**Table 3. RMA regression for 2D over 4D.**

| | | | WOMEN | | MEN | | ALL | |
|---|---|---|---|---|---|---|---|---|
| | | | Coeff.±SEM | 95% BCI | Coeff.±SEM | 95% BCI | Coeff.±SEM | 95% BCI |
| RIGHT HAND | Full sample | SLOPE | 1.007 [NS] ±0.019 | (0.970, 1.043) | 1.074** ±0.019 | (1.036, 1.111) | 1.104** ±0.011 | (1.080, 1.125) |
| | | INTERCEPT | 0.129[NS] ±0.127 | (-0.123, 0.391) | -0.125[NS] ±0.144 | (-0.529, 0.030) | -0.503[++] ±0.082 | (-0.657, -0.336) |
| | | | $R^2$ = 0.698 | | $R^2$ = 0.701 | | $R^2$ = 0.805 | |
| | Sub-sample | SLOPE | 1.037 [NS] ±0.027 | (0.984, 1.090) | 1.062* ±0.027 | (1.009, 1.112) | 1.105** ±0.016 | (1.073, 1.135) |
| | | INTERCEPT | -0.072[NS] ±0.187 | (-0.429, 0.298) | -0.167[NS] ±0.234 | (-0.541, 0.232) | -0.511[++] ±0.116 | (-0.736, -0.282) |
| | | | $R^2$ = 0.674 | | $R^2$ = 0.687 | | $R^2$ = 0.796 | |
| LEFT HAND | Full sample | SLOPE | 1.019 [NS] ±0.018 | (0.986, 1.053) | 1.061* ±0.020 | (1.021, 1.101) | 1.077** ±0.011 | (1.055, 1.099) |
| | | INTERCEPT | 0.059 [NS] ±0.125 | (-0.172, 0.287) | -0.176[NS] ±0.154 | (-0.485, 0.122) | -0.314[++] ±0.080 | (-0.470, -0.160) |
| | | | $R^2$ = 0.713 | | $R^2$ = 0.713 | | $R^2$ = 0.821 | |
| | Sub-sample | SLOPE | 1.050 [NS] ±0.026 | (0.997, 1.097) | 1.092* ±0.034 | (1.023, 1.156) | 1.086** ±0.017 | (1.051, 1.119) |
| | | INTERCEPT | -0.143[NS] ±0.179 | (-0.470, 0.218) | -0.418[NS] ±0.254 | (-0.896, 0.094) | -0.384[+] ±0.120 | (-0.620, -0.135) |
| | | | $R^2$ = 0.704 | | $R^2$ = 0.684 | | $R^2$ = 0.808 | |

We used t-test (2-tailed) to test if the slopes differ from 1 (** p<0.001; * p<0.05; [NS] p>0.05) and if intercepts are different from 0 ([++] p<0.001; [+] p<0.05; [NS] p>0.05). Full sample includes 895 females and 934 males for the right hand and 899 females and 775 males for the left hand. The subsample of published data includes 474 males and 471 females for the right hand and 327 males and 474 females for the left hand. BCI: Bootstrap confidence intervals. The 95% bootstrap confidence intervals were computed with 1999 rearrangements.

**Table 4. 2D:4D values according to sexual orientation categories for men and women.**

|  |  | Right 2D:4D | | Left 2D:4D | |
|---|---|---|---|---|---|
|  |  | **Men** | **Women** | **Men** | **Women** |
| SexOrCateg | Homosexual | 0.982 ± 0.005 (32) | 0.968 ± 0.008 (14) | 0.980 ± 0.007 (24) | 0.974 ± 0.010 (14) |
|  | Bisexual | 0.979 ± 0.009 (12) | 0.978 ± 0.007 (33) | 0.982 ± 0.007 (11) | 0.974 ± 0.005 (33) |
|  | Heterosexual | 0.960 ± 0.002 (345) | 0.973 ± 0.002 (454) | 0.964 ± 0.002 (272) | 0.972 ± 0.001 (455) |
|  |  | $F_{2,386} = 10.118$, p<0.001 | $F_{2,498} = 0.470$, p = 0.625 | $F_{2,304} = 4.363$, p = 0.014 | $F_{2,499} = 0.112$, p = 0.894 |
| ESOC | Homosexual | 0.977 ± 0.003 (75) | 0.969 ± 0.007 (24) | 0.974 ± 0.004 (67) | 0.970 ± 0.007 (24) |
|  | Bisexual | 0.974 ± 0.006 (34) | 0.978 ± 0.003 (116) | 0.978 ± 0.006 (34) | 0.974 ± 0.003 (117) |
|  | Heterosexual | 0.962 ± 0.001 (594) | 0.975 ± 0.001 (748) | 0.963 ± 0.001 (520) | 0.973 ± 0.001 (751) |
|  |  | $F_{2,700} = 9.683$, p<0.001 | $F_{2,885} = 0.736$, p = 0.479 | $F_{2,618} = 6.005$, p = 0.003 | $F_{2,889} = 0.154$, p = 0.858 |

Values are mean ± SEM. N appears in each cell in parentheses.

residuals obtained from the left-hand RMA regression between women (-1.694 ± 0.783 $10^{-2}$) and men (1.965 ± 0.938 $10^{-2}$). In other words, the fourth finger in women is shorter than expected according to the length of the second finger, while the opposite situation occurs in men. There is a significant and negative correlation between the residual value and the length of the second finger for both the right ($r_{1829}$ = -0.227, p <0.001) and the left hand ($r_{1674}$ = -0.217, p <0.001). The correlation coefficients of the fourth finger with the residuals are of the same value but with the opposite sign, as expected for RMA regressions. This results indicates that for large fingers 2 we find finger 4 smaller than expected, and that for small finger 2 finger 4 is longer than expected.

To analyse the relationship between sexual orientation and 2D:4D, we performed analyses with both discrete and continuous sexual orientation measures, as well as with the ESOC variable. The results for the sexual orientation measures that include categories are shown in Table 4. The correlation between continuous sexual orientation and 2D:4D was not significant in women neither for the right hand ($\rho_{439}$ = -0.010, p = 0.830) nor for the left hand ($\rho_{442}$ = 0.005, p = 0.921). In men, the correlation was significant with 2D:4D of the left hand ($\rho_{340}$ = -0.140, p = 0.010) but not with 2D:4D of the right hand ($\rho_{340}$ = -0.057, p = 0.292). Regression coefficients were significantly different between men and women for the left hand data (Z = 2.015, p = 0.044) but not for the right hand (Z = 0.649, p = 0.516). These results are quite similar when only Spaniards are included in the analyses.

We analysed if there were differences in 2D:4D according to the declared handedness (Table 5). We also tested the correlations between 2D:4D of each hand and dHGS. Considering women and men jointly, there were no significant correlations neither for the right hand ($\rho_{1187}$ = 0.015, p = 0.599), the left hand ($\rho_{1190}$ = 0.009, p = 0.765), nor the difference of 2D:4D

**Table 5. 2D:4D values according to handedness categories for men and women.**

|  | Right Hand | | Left Hand | | Difference (r-l) | |
|---|---|---|---|---|---|---|
|  | **Women** | **Men** | **Women** | **Men** | **Women** | **Men** |
| Right handed | 0.977±0.002 (380) | 0.961±0.001 (681) | 0.973±0.002 (383) | 0.964±0.001 (681) | 3.934±1.159 $10^{-3}$ (380) | -2.795±0.948 $10^{-3}$ (680) |
| No-right handed | 0.978±0.005 (34) | 0.965±0.004 (94) | 0.978±0.005 (34) | 0.967±0.003 (94) | 0.358±4.113 $10^{-3}$ (34) | -1.067±2.338 $10^{-3}$ (94) |
|  | $t_{412}$ = -0.259, p = 0.796 | $t_{773}$ = -1.148, p = 0.251 | $t_{415}$ = -0.889, p = 0.374 | $t_{773}$ = -0.633, p = 0.527 | $t_{412}$ = -0.380, p = 0.408 | $T_{772}$ = 0.521, p = 0.495 |

Values are mean ± SEM. N appears in each cell in parentheses. Difference is the difference in the 2D:4D of the hands (2D:4D on the right minus 2D:4D on the left).

**Table 6. Distribution of participants according to handedness and sexual orientation categories for men and women.**

| | | dHGS | | % right handed | |
| --- | --- | --- | --- | --- | --- |
| | | **Men** | **Women** | **Men** | **Women** |
| SexOrCateg | Homosexual | 0.100 ± 0.022 (24) | -0.119 ± 0.024 (2) | 95.8% (24) | 0.00% (2) |
| | Bisexual | 0.010 ± 0.046 (11) | 0.059 ± 0.043 (7) | 90.9% (11) | 100.0% (7) |
| | Heterosexual | 0.054 ± 0.007 (272) | 0.071 ± 0.015 (43) | 90.4% (272) | 90.7% (43) |
| | | $\chi^2_2 = 5.061$, p = 0.080 | $\chi^2_2 = 4.597$, p = 0.100 | $\chi^2_2 = 0.774$, p = 0.679 | $\chi^2_2 = 16.457$, p<0.001 |
| ESOC | Homosexual | 0.073 ± 0.015 (67) | 0.099 ± 0.040 (11) | 86.6% (67) | 63.6% (11) |
| | Bisexual | 0.042 ± 0.022 (34) | 0.058 ± 0.012 (87) | 94.1% (34) | 88.5% (87) |
| | Heterosexual | 0.060 ± 0.005 (520) | 0.074 ± 0.006 (318) | 88.3% (521) | 93.7% (318) |
| | | $\chi^2_2 = 1.098$, p = 0.577 | $\chi^2_2 = 1.738$, p = 0.419 | $\chi^2_2 = 1.311$, p = 0.519 | $\chi^2_2 = 14.430$, p = 0.001 |

Values are mean ± SEM. N appears in each cell in parentheses. Comparisons between the categories of sexual orientation were performed using Kruskall-Wallis test.

between hands ($\rho_{1186} = 0.012$, p = 0.672). In men there were no significant correlations for right ($\rho_{773} = -0.038$, p = 0.294) or left 2D:4D ($\rho_{773} = -0.015$, p = 0.668). In women we also did not find a significant correlation between dHGS and 2D:4D of the right hand ($\rho_{414} = 0.086$, p = 0.080) or with the left hand ($\rho_{414} = 0.022$, p = 0.654). Regression coefficients were significantly different between men and women for the right-hand data (Z = 2.034, p = 0.042), but not for the left-hand data (Z = 0.606, p = 0.545).

Finally, we analysed to what extent handedness and sexual orientation are related. The correlation between dHGS and OrSexCont is not significant neither in women ($\rho_{416} = -0.016$, p = 0.752) nor in men ($\rho_{341} = 0.057$, p = 0.293), and these coefficients are not different from each other (Z = 0.996, p = 0.319). There are also no significant differences in men in their continuous measure of sexual orientation (U = 6935.5, $N_1 = 291$, $N_2 = 50$, p = 0.591) between right-handed (0.801 ± 0.017) and non-right-handed (0.794 ± 0.044). However, in women there are significant differences (U = 4749.0, $N_1 = 382$, $N_2 = 34$, p = 0.008) in the continuous measure of sexual orientation between right-handed (0.836 ± 0.010) and non-right-handed (0.733 ± 0.042). In this sense, there are significant differences in the proportion of self-declared right-handed women between the different categories of sexual orientations considered in the OrSexCont variable and in the ESOC variable (see Table 6). On the other hand, there are no significant differences in dHGS when comparing these same categories (Table 6).

## Discussion

Our results show that the sex differences in 2D:4D are not exclusively due to differences in size. While the length of the fingers accounts for 2.2% of the variance in 2D:4D ($R^2$, Table 2) sex represents 4.3% of the variance in 2D:4D (estimated from Cohen's d, Table 1). An approach to easily explain the importance of size in the 2D:4D difference between sexes is to compare the actual difference with the expected differences based on the mean size of the fingers of each sex. The differences in 2D:4D between the sexes are $1.362 \ 10^{-2}$ for the right hand and $0.844 \ 10^{-2}$ for the left hand (Table 1). We can estimate the 2D:4D from the average size of the fingers of each sex (Table 1) using the OLS regressions developed including both sexes (Table 2). The differences between the estimated 2D:4D for males and females are $0.625 \ 10^{-2}$ for the right hand and $0.421 \ 10^{-2}$ for the left. We can also calculate the 2D:4D values using the consensus equation obtained from the OLS regressions performed for each sex independently (table notes in Table 2). The differences in 2D:4D estimated with this method for the mean size of the finger length of each sex are $0.113 \ 10^{-2}$ for the right hand and $0.112 \ 10^{-2}$ for the left.

In both cases, the estimated differences are less than half the actual difference. Although it must be taken into account that size and sex are not independent variables in humans when interpreting this data, it seems that 2D:4D differences between the sexes are not just a consequence of size differences, contrary to the observed in some previous results [37, 38]. In this sense, we think that 2D:4D continues to be an adequate indicator of prenatal exposure to sex hormones, since this is the main mechanism behind the morphological differences between females and males. This does not exclude other sources of variability that also affect sex differences in 2D:4D [29, 128], as recently concluded by other studies [21], making it difficult to attribute its individual variability solely to the effect of prenatal sex hormone. We must take into consideration that only 61.8% of the times we take a woman and a man at random, she would have a value of right 2D:4D higher than his. This indicates that 2D:4D requires very large sample sizes to produce significant results, and is far from being a good indicator of individual hormone exposure during fetal development. However, it is still the best indicator considering the difficulties in collecting any other sex differences linked to prenatal hormones exposure in large samples.

Our findings indicate that the relationship between the length of the second and fourth finger is rather isometric than allometric, with many results in favor of isometry and a few of them showing allometry. OLS regressions mainly indicate isometry, but this can also be concluded from some results with RMA regressions. Interestingly, the analyses indicate isometry when the sexes are considered separately, but not when they are analysed together. This suggests the relationship of finger sizes may be different in men and women, which is partially supported by other results. First, in RMA regression comparing the lengths of fingers 2 and 4 we obtain slopes greater than 1 in men, but not in women, although in both situations the value is close to one (even when only in one case the slopes are significantly different when we compare them between sexes). Second, the differences between the sexes in the residuals obtained with all the regressions carried out without separating by sex. Thus, although the difference must be very small when we are unable to clearly detect it with a sample size of almost 2000 people, we are inclined to think that the relationship between 2D and 4D lengths is somewhat different according to sex. In any case, our data indicate that the size relationship approaches the limit between isometry and allometry, with regressions slopes equivalent to those observed in other recent studies [21, 121].

Our results regarding sexual orientation and 2D:4D in men are in line with some reported for different populations [36, 88–92], in which men who are categorized as non-heterosexual show more feminine 2D:4D. However, our results are contrary to what was previously observed for the self-defined as white population [36]. The results support previously described effects of ethnicity on the relationship between 2D:4D and sexual orientation [36, 85, 129], even within Caucasian populations [88]. Our sample is mainly composed of self-reported white Spaniards (92.53%), genetically related to North Africans, Portuguese, Italians, French and Irish [130, 131]. Subjects in our sample from Latin-American countries who consider themselves as white are probably genetically related to these same populations. Thus, we consider that there are no major ethnic differences within our sample (even so, analysing only Spaniards we obtained the same results). Of course, the proportion of sexual orientation categories by sex in our sample of Spaniards is in line with those described in the previous bibliography. We attribute the differences observed with other samples of European populations to differences in ethnic background, which also causes 2D:4D differences between different ethnicities and nationalities [103, 111, 132]. We also attribute to ethnic differences that we found no relationship between 2D:4D and sexual orientation in women. Previous meta-analyses had indicated that non-heterosexual women show significant lower 2D:4D values than heterosexual women, and that this relationship is more strongly observed in European populations [36].

However, in an ethnically homogeneous sample of Finns the values obtained for non-heterosexual women indicated higher values of 2D:4D [88]. Everything seems to indicate that a fine control of ethnicity, beyond the simple self-report in one of the usual considered categories (White, Black, Hispanic, Asian. . .), is necessary in studies that involve 2D:4D. Controlling genetic variation of participants has allowed to obtain clearer and more robust results in recent studies [133], leading us to postulate that the weak control of these variability can explain the mixed results obtained when analysing different aspects of 2D:4D in heterogeneous samples, including its weakest or strongest link to body size. It is impossible, for economic and technical reasons, to analyse in depth the genetic background of each participant in studies interested in analysing the adult effect of exposure to sex hormones during fetal development. We suggest asking informative questions about ancestry, such as grandparents' home country, for a slightly more precise characterization of individuals' biological ancestry.

We could not find any significant relationship between 2D:4D and handedness, measured neither as preference nor as difference in hand grip strength. This is not completely unexpected given the weak relationships found in the meta-analysis [24]. To detect the effect described in the meta-analysis for the self-reported handedness preference we would have needed a sample of more than 30,000 subjects. Although studies that measured handedness with a task indicated that it would be easier to detect a relationship, it is also true that most studies have less than one hundred subjects (Stoyanov et al. 2009 [102] N = 36, Beaton et al. [134] 2011 N = 40, Fink et al. 2004 [104] N = 93, Beaton et al. 2012 [105] N = 68; except Manning et al. 2000 [101], N = 406). While our sample far exceeds these sample sizes (N = 1189), and although our measure of handedness on a task fits perfectly with other measures of preference and performance, this does not imply that hand performance is not related with 2D:4D at all. One possible difficulty in finding a relationship is related to the measurement of a single task, and perhaps a mixed index including several tasks would be a more accurate measure of performance [96, 97]. Another difficulty may simply be the requirement of sample sizes equivalent to those needed to detect a relationship with handedness measured as a preference.

Finally, in the analysis of homosexuality and handedness measured as a preference we found the expected relationship in women, with more left-handers among non-heterosexuals [108]. Although the results with handedness performance are not significant, the values we obtained for bisexual women do not seem as extreme as for self-described homosexuals. They show a mixed handedness performance, as proposed in some recent studies [74, 78, 109, 110]. Our sample of non-heterosexual and non-right-handed subjects is smaller than those in which participants are selected for their sexual orientation or handedness. The size of the sample logically makes it difficult to find an effect, but using an unselected sample has fewer problems with selection bias [78], especially if we take into account that our sample was not recruited to specifically study the relationships between these variables. In men we found no relationship between preferred handedness and homosexuality, as reported in some previous studies [78, 109, 113, 114]. This can be explained taking into account recent results suggesting that biological features during development influence same-sex sexual orientation in men only in specific subgroups [106]. Handedness and sexual orientation seem to be consistently associated in only part of the analysed sample (around 15%), a split made by following profiles based on similarities between individuals in well-established biomarkers of sexual orientation. This makes it difficult to find a clear relationship between the two characteristics in the whole sample. In this sense, our results regarding handedness contrast with those that we obtained with 2D:4D, in which its relationship with sexual orientation in men is clearer. Although both traits are associated with prenatal exposure to hormones, many other factors influence them, such as genetic, immunological or environmental differences [109]. In our opinion, and considering all the

limitations exposed above, 2D:4D is a trait more linked to prenatal sex hormone levels than handedness, which seems to be more easily modified by learning and cultural environment.

Regarding handedness performance, although the results are not significant, it seems that men who define themselves bisexual present handedness values that indicate greater ability with the left hand than homosexuals. Among homosexual men, we found performance values that indicate a more skillful use of the right hand, perhaps in line with the proposition of more extreme values of handedness, skewed to the right or to the left, in homosexual men [115, 116]. A larger study with handedness performance measured simultaneously with a continuous measurement of homosexuality will help clarifying the possible sex difference in the relationship between handedness and sexual orientation. We expected that measuring both sexual orientation and handedness with continuous variables, avoiding bimodality problems when using measures with categories [93], would allow us to detect the proposed relationships that need to distinguish between mild and extreme values in both variables. Nevertheless, measuring handedness with a continuous variable has not made it easier for us to detect relationships with 2D:4D, in the same way that the use of a continuous variable has not allowed us to further refine the relationship between 2D:4D and sexual orientation [85, 92]. Even when continuous measures fit well in our sample with categories when measured simultaneously, further studies are needed to understand how individuals translate from a continuing trend to self-categorization.

Our main conclusion is that 2D:4D presents differences between the sexes beyond the issues associated with differences in body size. Likewise, in our opinion, 2D:4D can still be used as an indicator of intrauterine exposure to sex hormones if some precautions are taken. First, the required sample size must be very large. This is clearly demonstrated by our inability to detect the previously described 2D:4D association with handedness, despite our relatively large sample size. It should also be noted that we have been able to observe in our sample associations with previously described aspects, such as sexual orientation, but which seem to depend on the ethnicity of the sample. This establishes a second suggestion to improve any study that considers 2D:4D as a variable: the sample must present a certain homogeneity of genetic background, or include a measure of genetic kinship more accurate than self-reported ethnicity, such as considering ancestry. We are aware that it is not always feasible to meet these conditions, so we suggest at least that any study considering 2D:4D make every effort to take care of the genetic background to allow the development of appropriate future meta-analyses. In addition, to facilitate further meta-analysis, we also suggest making raw finger measurements available.

## Supporting information

**S1 File. Dataset.** Finger lengths, sexual orientation and handedness measures of the 1835 participants.
(XLSX)

## Author Contributions

**Conceptualization:** Denisa Cristina Lupu, Enrique Turiegano.

**Data curation:** Denisa Cristina Lupu, Claudia Rodriguez-Ruiz, Miguel Pita, Enrique Turiegano.

**Formal analysis:** Denisa Cristina Lupu, Ignacio Monedero, Enrique Turiegano.

**Funding acquisition:** Enrique Turiegano.

**Investigation:** Denisa Cristina Lupu, Ignacio Monedero, Enrique Turiegano.

**Methodology:** Denisa Cristina Lupu, Ignacio Monedero, Claudia Rodriguez-Ruiz, Miguel Pita, Enrique Turiegano.

**Software:** Ignacio Monedero, Enrique Turiegano.

**Writing – original draft:** Denisa Cristina Lupu, Ignacio Monedero, Claudia Rodriguez-Ruiz, Miguel Pita, Enrique Turiegano.

**Writing – review & editing:** Denisa Cristina Lupu, Ignacio Monedero, Claudia Rodriguez-Ruiz, Miguel Pita, Enrique Turiegano.

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
