## [Decision Letter · Decision Letter 0]

21 Nov 2022

PONE-D-22-03299The case against 2D:4D: more data about its conflicting results on handedness, sexual orientation and sex differences.PLOS ONE

Dear Dr. Turiegano,

Thank you for submitting your manuscript to PLOS ONE. After careful consideration, we feel that it has merit but does not fully meet PLOS ONE’s publication criteria as it currently stands. Therefore, we invite you to submit a revised version of the manuscript that addresses the points raised during the review process.

Your manuscript has been assessed by two peer-reviewers and their reports are appended below. 

The reviewers comment that some aspects of the manuscript would benefit from additional explanation or clarification, and that the title of the study is confusing. In addition, the reviewers have requested some clarification regarding aspects of the statistical analysis described in this study. 

Could you please revise the manuscript to carefully address the concerns raised?

We look forward to receiving your revised manuscript.

Kind regards,

Maria Elisabeth Johanna Zalm, Ph.D

Editorial Office

PLOS ONE

Journal Requirements:

“NO authors have competing interests”

“The experiments that allowed us to collect the analised data were supported by the Spanish Ministry for Science and Innovation [grant numbers: BFU2010-10981-E; ECO2015-66281-P; PID2019-105895GB-I00], the Ministry of Economics and Competitiveness [ECO2011-28750; ECO2012-33243] and by Department of Biology (UAM) Research Funds [BIOUAM05-2019].”

‘ET- Spanish Ministry for Science and Innovation [grant numbers: BFU2010-10981-E; ECO2015-66281-P; PID2019-105895GB-I00]. (https://www.ciencia.gob.es/)

ET- Ministry of Economics and Competitiveness [ECO2011-28750; ECO2012-33243] (https://portal.mineco.gob.es/)

ET- Department of Biology (UAM) Research Funds [BIOUAM05-2019]. (https://www.uam.es/Ciencias/DBIO)”

Reviewers' comments:

Reviewer's Responses to Questions

**Comments to the Author**

1. Is the manuscript technically sound, and do the data support the conclusions?

Reviewer #1: Yes

Reviewer #2: Yes

2. Has the statistical analysis been performed appropriately and rigorously? 

Reviewer #1: Yes

Reviewer #2: Yes

3. Have the authors made all data underlying the findings in their manuscript fully available?

Reviewer #1: Yes

Reviewer #2: Yes

4. Is the manuscript presented in an intelligible fashion and written in standard English?

Reviewer #1: Yes

Reviewer #2: Yes

5. Review Comments to the Author

Reviewer #1: The authors address a long-debated issue in the field – whether sex differences in 2D:4D is simply due to allometry or a result of prenatal androgen exposure. The authors thoroughly and fairly describe the state of the literature, and address many of the limitations of prior work by using a large sample to test the primary criticism against the use of 2D:4D (i.e., allometry). The authors find that sex differences in 2D:4D cannot be explained by allometry alone, and thus suggest that 2D:4D is a valid proxy for prenatal androgens, provided certain precautious are met. These considerations for future research are outlined (i.e., using large sample sizes and considering ethnicity beyond self-report – i.e., asking grandparents home country). As such, I recommend this manuscript be accepted for publication with minor revisions, as outlined below.

---------------

Point-by-point comments:

Title is confusing – it reads as though this will be an article with more conflicting data, and the data will be largely against the use of 2D:4D.

Abstract:

“The debate has also questioned..” this is the first mention of debate, so I would omit “also”. I would rephrase to something along the lines of: “However, this proxy has been met with criticism, especially in relation to other purported features associated with prenatal androgen such as handedness and sexual orientation.”

Introduction:

“considerable controversy over the past two years..”

its been much longer; for example, see: Kratochvíl, L., & Flegr, J. (2009). Differences in the 2nd to 4th digit length ratio in humans reflect shifts along the common allometric line. Biology letters, 5(5), 643-646.

Line 142 “sexual differences” – do you mean sex differences or sexual orientation differences?

Line 179 “posterior studies” – previous?

Methods:

Line 233 –“analised” analysed?

Please provide more details on how measurements of digits were obtained (e.g., middle/centre of tip to palm?)

Results/Discussion

Could some of the variance due to allometry be accounted for by the role of androgens in body size? i.e., androgens contribute to sex differences in body size and 2D:4D, so some shared variance would be expected?

It may be worth noting that prenatal androgens may only lead to same-sex sexual orientation among a subset of individuals (i.e., Swift-Gallant et al., 2019; PNAS finds that handedness only explains sexual orientation among <6% of non-heterosexual men; also see recent review by VanderLaan et al., 2022, Archives of Sex Beh), and so a relationship between these proxies for prenatal androgen and sexual orientation may not be found when looking at all gay men/non-heterosexual as one group.

Reviewer #2: I have reviewed the manuscript and I think its well written and discussed. I do not have any issues with the title, abstract, introduction and discussion. I however, will like to recommend minor revisions in the methods and results section

6. PLOS authors have the option to publish the peer review history of their article (what does this mean?). If published, this will include your full peer review and any attached files.

Reviewer #1: No

Reviewer #2: **Yes: **MOSES BANYEH

---

## [Author Response · Author response to Decision Letter 0]

16 Dec 2022

Dear Dr. Maria Elisabeth Johanna Zalm, Editorial Board Member of PLoS One,

Thank you for giving us the opportunity to review and resubmit our manuscript currently entitled “In support of 2D:4D: more data exploring its conflicting results on handedness, sexual orientation and sex differences.” to PLoS One.

The comments made by the reviewers have been really useful for improving our manuscript. We have taken the vast majority of the comments on board, making modifications that are detailed in our response to the reviewers. When the comment was not taken on board, we offered an explanation as to why that was the case. We have modified the title to improve its clarity. We have also rewritten some of the paragraphs throughout the manuscript to provide more comprehensible and clear information, taking into account the perspective provided by the suggested references. 

In compliance with the journal requirements please note that:

- The authors declare that no competing interests exist. We state "The authors have declared that no competing interests exist." Please, change the online submission form on our behalf.

- We have removed the funding-related text from the manuscript. The Funding Statement should read as follows: 

“‘ET- Spanish Ministry for Science and Innovation [grant numbers: BFU2010-10981-E; ECO2015-66281-P; PID2019-105895GB-I00]. (https://www.ciencia.gob.es/)

ET- Ministry of Economics and Competitiveness [ECO2011-28750; ECO2012-33243] (https://portal.mineco.gob.es/)

ET- Department of Biology (UAM) Research Funds [BIOUAM05-2019]. (https://www.uam.es/Ciencias/DBIO)”. 

Please, change the online submission form on our behalf.

- We have reviewed the reference list to ensure that it is complete and correct. We cited an article that included two published errata, but the article currently available online already includes this small modifications (without changing its ISBN). We have also removed an article from the reference list because it was not published in an appropriate journal, and the information contained in it was also included in other cited articles. 

Finally, we asked some English-speaking colleagues to proofread the manuscript, and we have formatted it according to the PLoS One requirements. As a result of all these changes, the manuscript has definitely gained clarity and scope. We are very grateful to you and the reviewers for your suggestions and encouragement.

Yours sincerely, 

Enrique Turiegano

Dear Reviewers, 

First of all, we would like to thank you very much for your report.

We found your comments to be very thoughtful. We have carefully reviewed the manuscript and addressed the issues you raised by including appropriate discussions. We also corrected the typos. We sincerely believe that the current version of the manuscript is more accurate, robust and readable. Thank you again for the time and effort you have put into reviewing our manuscript.

Below you will find detailed answers to each point you mentioned. When the comment does not lead us to modify the manuscript, we offer an explanation of why this is so.

Thank you very much.

Reviewer #1: The authors address a long-debated issue in the field – whether sex differences in 2D:4D is simply due to allometry or a result of prenatal androgen exposure. The authors thoroughly and fairly describe the state of the literature, and address many of the limitations of prior work by using a large sample to test the primary criticism against the use of 2D:4D (i.e., allometry). The authors find that sex differences in 2D:4D cannot be explained by allometry alone, and thus suggest that 2D:4D is a valid proxy for prenatal androgens, provided certain precautious are met. These considerations for future research are outlined (i.e., using large sample sizes and considering ethnicity beyond self-report – i.e., asking grandparents home country). As such, I recommend this manuscript be accepted for publication with minor revisions, as outlined below.

---------------

Point-by-point comments:

Title is confusing – it reads as though this will be an article with more conflicting data, and the data will be largely against the use of 2D:4D.

We have changed the title, in an attempt to minimize possible confusion about the role of our results supporting the utility of 2D:4D. The exact placements of the changes are lines 1-2.

Abstract:

“The debate has also questioned..” this is the first mention of debate, so I would omit “also”. I would rephrase to something along the lines of: “However, this proxy has been met with criticism, especially in relation to other purported features associated with prenatal androgen such as handedness and sexual orientation.”

We have introduced changes to the manuscript in order to clarify this part of the abstract. The exact placements of the changes are lines 14-16.

Introduction:

“considerable controversy over the past two years..”

its been much longer; for example, see: Kratochvíl, L., & Flegr, J. (2009). Differences in the 2nd to 4th digit length ratio in humans reflect shifts along the common allometric line. Biology letters, 5(5), 643-646.

We have introduced changes to the manuscript reinforcing the idea that the controversy has existed for a long time, but in the two last years it has strengthened. We do not present all the considered citations questioning 2D:4D in this part of the introduction for ease of reading, but we mention them below in the introduction, including the mentioned paper of Kratochvíl and Flegr (cite [37]). The exact placements of the changes are lines 48-50. 

Line 142 “sexual differences” – do you mean sex differences or sexual orientation differences?

We have corrected the typo along the manuscript. The exact placements of the changes are lines 39, 82, 118, 140 and 366. 

Line 179 “posterior studies” – previous?

We have corrected the typo in the manuscript. The exact placements of the change is line 180.

Methods:

Line 233 –“analised” analysed?

We have corrected the typo in the manuscript. The exact placements of the change are lines 174 and 234.

Please provide more details on how measurements of digits were obtained (e.g., middle/centre of tip to palm?)

We have updated the Methods section by detailing how the finger lengths were measured. The exact placements of the changes are lines 241-242.

Results/Discussion

Could some of the variance due to allometry be accounted for by the role of androgens in body size? i.e., androgens contribute to sex differences in body size and 2D:4D, so some shared variance would be expected?

2D:4D is affected by sex differences, but also by genetic factors, some of which must be responsible for the effect of ethnicity on the character. On the other hand, although the physical and physiological differences between the sexes fundamentally depend on the different exposure to sex hormones, hormones act on the phenotype through changes in gene expression. This implies that genetic differences between individuals, as well as the interaction of gene expression with the environment, influence how the final phenotype is reached. That is, there are many factors to affirm with certainty that the variation between individuals in 2D:4D is mainly due to the effect of androgens. Of course, some variance will be shared between other features related to hormonal exposure, such as body size, and 2D:4D. But we think that the relationship will be clouded by a multitude of factors. We have modified a paragraph in the discussion exposing these difficulties. The exact location of the change is lines 472-482.

It may be worth noting that prenatal androgens may only lead to same-sex sexual orientation among a subset of individuals (i.e., Swift-Gallant et al., 2019; PNAS finds that handedness only explains sexual orientation among <6% of non-heterosexual men; also see recent review by VanderLaan et al., 2022, Archives of Sex Beh), and so a relationship between these proxies for prenatal androgen and sexual orientation may not be found when looking at all gay men/non-heterosexual as one group.

This is a very relevant and significant observation. It is possible that our inability to find a relationship between handedness and sexual orientation in males is due to this phenomenon in which the effect of the biological variable is only seen in a subsample of men. In this sense, it is interesting that we do find a clear relationship between 2D:4D and sexual orientation in men. It is true that both variables, handedness and 2D:4D, are related to the levels of sex hormones during development, but they are also influenced by many other aspects of our biology such as genetic, immunological or environment-related differences during development (for instance, consider the effect of developmental instability on handedness). Therefore, it is not uncommon for their relationship to sexual orientation to be different. We have added a paragraph on this topic in the discussion section. The exact placement of the changes are lines 175-178 and lines 554-565.

Reviewer #2: Thank you for the opportunity to review this manuscript. The authors sought to test whether sexual dimorphism in the 2D:4D is due to allometric or isometric changes in the human fingers and whether the 2D:4D can represent prenatal hormone exposure. They further sought to determine whether the 2D:4D has an impact on sexual orientation and handedness in both adult males and females. It was observed that the 2D:4D may be used as an index of prenatal hormone exposure, but with caution. Moreover, the 2D:4D was associated with sexual orientation in only males but no association with handedness was detected. The manuscript is well written and well discussed. I will recommend its publication if the authors can address these minor revisions: 

Methods/results 

1. Line 282-284: “Based on the data obtained in the population for which we had both measures, we decided to consider those subjects with values greater than 0.7 as heterosexual, those with values below 0.3 as homosexual, and those with values between 0.3 and 0.7 as bisexual”. What was the scientific basis for your decision?

We have added an explanation as to why we used that rule to classify the subject in different sexual orientation categories. The exact placements of the changes are lines 286-291. Our classificatory method considers an equal size range of 0.3 to categorize participants as heterosexual or homosexual, as it is usually done to categorize continuous variables that measure sexual orientation (see ref. [8]). Our classification method maximizes the number of properly classified subjects, since ESOC correctly classifies 96.2% of the sample whose sexual orientation was measured with both continuous and categorical variables.

2. Line 334-335:” The slope is not significantly different from 0 except in the regression fitted for the right-hand data for the men in the full sample”. Line 386: “We used t-test to test if the slopes are different from 0”. Line 398: “We used a t-test to test if the slopes differ from 1” Did you use a 1-tailed or 2-tailed test and at what confidence interval?

We have clarified throughout the manuscript that each t-test used is 2-tailed. The exact placements of the changes are lines 387 and 399. We included in tables 2 and 3 the Bootstrap confidence intervals for each computed slope and intercept. In order to provide the most useful information possible without making the tables too large, we have preferred not to change these confidence intervals in the tables. Any reader interested in the usual confidence intervals can calculate them with the mean and the SEM (provided in each table for each slope and intercept) simply by adding and subtracting from the mean the product of SEM multiplied by the z-score associated with the desired confidence level (1.96 for a 95% confidence level and this large sample size).

3. Table 1-6. You summarized the results as mean (SD) in Table 1 but mean (SEM) in Tables 4-6. What informed this decision and why were you not consistent? 

It was a typo. The values included in table 1 are also SEM. It has been corrected in line 333.

4. Table 1: It might be useful to define the effect size in the table legend as; Cohens d: negligible (d<0.20), small (0.20≤d<0.50), medium (0.50≤d≤0.80) and large (d>0.80).

We have introduced the suggested change in the manuscript. The exact placement of the sentence is lines 335-337.

5. Table 5: The difference in the 2D:4D is traditionally determined as the right-left difference (Dr-l). A lower Dr-l may indicate increased exposure to prenatal androgens as lower righthand 2D:4D is associated with masculine traits.

We have modified the way of calculating this difference and we have clarified it in the manuscript (line 433). We have computed the difference as right minus left. The sign changes of the 2D:4D difference values have been modified in the manuscript. The exact placements of the changes are in the last two columns of table 5 (lines 431-433)

---

## [Editor Report · Decision Letter 1]

2 Jan 2023

In support of 2D:4D: more data exploring its conflicting results on handedness, sexual orientation and sex differences.

PONE-D-22-03299R1

Dear Dr. Turiegano,

We’re pleased to inform you that your manuscript has been judged scientifically suitable for publication and will be formally accepted for publication once it meets all outstanding technical requirements. Also, please see Editor comments below for a few minor suggested edits.

Kind regards,

Ashlyn Swift-Gallant

Guest Editor

PLOS ONE

Additional Editor Comments (optional):

Below, I offer a few minor edits to consider in your proofs. Also, to preserve transparency in the review process, I must divulge that I was recently invited to serve as a guest editor and previously served as a reviewer. Please contact staff editor Dr. Maria Elisabeth Johanna Zalm if you have any concerns in regards to the handling of your manuscript.

Suggested Edits:

Abstract: line 21 – either remove“..the usually found..” or replace with “..the typically observed sex difference in this trait..”

Introduction: It will be 2023 when this manuscript is in press; thus, references 15 and 16 will not be in the last two years (2019, 2020). As such, please revise “last two years” to either last four or five years.

Line 104: change “…being the latter…” to “…the latter being..”

Line 149: change “…women who suffer CAH…” to “…women with CAH…”

Lines 216-218: remove “Indeed, some have been previously published, which minimizes self-selection sampling issues and observer bias problems.” – redundant.

Table 1: indicate units of measurement (cm?)

Line 514: “…non-heterosexual women show significant low 2D:4D values,…” please clarify whether you mean significantly lower than heterosexual women.

---

## [Editor Report · Acceptance letter]

17 Jan 2023

PONE-D-22-03299R1 

In support of 2D:4D: more data exploring its conflicting results on handedness, sexual orientation and sex differences. 

Dear Dr. Turiegano:

I'm pleased to inform you that your manuscript has been deemed suitable for publication in PLOS ONE. Congratulations! Your manuscript is now with our production department. 

Kind regards, 

on behalf of

Dr. Ashlyn Swift-Gallant 

Guest Editor

PLOS ONE